# The DNA damage response pathway regulates the expression of the immune checkpoint CD47

Lucy Ghantous [1,2], Yael Volman[1], Ruth Hefez[1], Ori Wald[1,3], Esther Stern[1], Tomer Friehmann[1], Ayelet Chajut[4], Edwin Bremer[5], Michal Dranitzki Elhalel [2,6✉] & Jacob Rachmilewitz [1,6✉]

CD47 is a cell surface ligand expressed on all nucleated cells. It is a unique immune checkpoint protein acting as "don't eat me" signal to prevent phagocytosis and is constitutively overexpressed in many tumors. However, the underlying mechanism(s) for CD47 overexpression is not clear. Here, we show that irradiation (IR) as well as various other genotoxic agents induce elevated expression of CD47. This upregulation correlates with the extent of residual double-strand breaks (DSBs) as determined by γH2AX staining. Interestingly, cells lacking mre-11, a component of the MRE11-RAD50-NBS1 (MRN) complex that plays a central role in DSB repair, or cells treated with the mre-11 inhibitor, mirin, fail to elevate the expression of CD47 upon DNA damage. On the other hand, both p53 and NF-κB pathways or cell-cycle arrest do not play a role in CD47 upregualtion upon DNA damage. We further show that CD47 expression is upregulated in livers harvested from mice treated with the DNA-damage inducing agent Diethylnitrosamine (DEN) and in cisplatin-treated mesothelioma tumors. Hence, our results indicate that CD47 is upregulated following DNA damage in a mre-11-dependent manner. Chronic DNA damage response in cancer cells might contribute to constitutive elevated expression of CD47 and promote immune evasion.

[1] Goldyne Savad Institute of Gene Therapy, Hadassah Medical Center, Faculty of Medicine, Hebrew University of Jerusalem, Jerusalem, Israel. [2] Department of Nephrology and Hypertension, Hadassah Medical Center, Faculty of Medicine, Hebrew University of Jerusalem, Jerusalem, Israel. [3] Department of Cardiothoracic Surgery, Hadassah Medical Center, Faculty of Medicine, Hebrew University of Jerusalem, Jerusalem, Israel. [4] Kahr Medical Ltd, Jerusalem, Israel. [5] Department of Hematology, University Medical Center Groningen, Groningen, The Netherlands. [6] These authors contributed equally: Michal Dranitzki Elhalel, Jacob Rachmilewitz. ✉email: michale@hadassah.org.il; rjacob444@gmail.com

SIRPα–CD47 pair of molecules has emerged as a novel immune checkpoint (IC) that targets the innate immune system. CD47 (also called integrin-associated protein, IAP) is a cell surface transmembrane glycoprotein widely expressed on many cells of epithelial and mesenchymal origin and is highly expressed on tumor cells[1]. SIRPα (also known as CD172a or SHPS-1) is a transmembrane glycoprotein receptor expressed predominantly on myeloid and neuronal cells and has been linked to cell adhesion[2,3]. SIRPα ligation by its cognate ligand CD47, used as a marker of 'self', results in a negative signal that inhibits phagocytic activity[3–6], therefore protecting CD47 expressing cells from elimination by engaging SIRPα on phagocytes. CD47 is consequently overexpressed on various types of tumors, allowing cancer cells to escape macrophage phagocytosis and immune surveillance[7–13]. Interestingly, CD47 upregulation was found to serve as a mechanism for leukemia progenitors to avoid phagocytosis[7,14]. Moreover, high CD47 expression is associated with poor prognosis in human malignancies[15,16].

The inhibitory checkpoint immunotherapy, which has revolutionized cancer treatment over recent years, aims to block these inhibitory signals and reactivate the immune system against the tumor. Indeed, blocking CD47-SIRPα interaction via anti-CD47 antibodies restored cancer cell phagocytosis and destruction by macrophages[11–14]. Surprisingly, CD47 blockade not only refurbished macrophage phagocytosis but also enabled T cell-mediated antitumor cytotoxic response[17,18]. Hence, these data suggest that CD47-SIRPα represents a new tumor escape mechanism that in addition to its ability to inhibit innate immunity also acts to restrain the adaptive immune response. Previously, we uncovered a unique mechanism wherein CD47 inhibits T cells indirectly by contact-dependent induction of aberrant antigen-presenting cell (APC) maturation. In turn, these CD47-conditioned APCs act as regulatory cells that actively regulate T-cell activity[19–22]. Of note, combination of CD47-SIRPα interaction blockade with ionizing radiation (IR) synergistically inhibits tumor growth[23–25].

Studies have suggested that cells may lose surface CD47 during apoptosis to enable phagocytic clearance (reviewed in[26]). Previously we demonstrated that while in apoptotic non-malignant cells CD47 surface expression is reduced at early stages of apoptosis, CD47 expression is significantly increased upon exposure to sub-lethal ionizing radiation (IR)[22]. In contrast, Vermeer et al., reported that irradiation reduced surface CD47 expression in head and neck squamous cell carcinoma in a dose-dependent manner[27]. Interestingly, in a series of studies David Roberts' group reported that blocking CD47 signaling induced by the CD47 ligand thrombospondin-1, or suppressing CD47 expression, protects normal tissues from IR injury while sensitizing the tumor cells[28–31], suggesting CD47 in tumor cells is important for DNA-damage repair. Notably, data implying that loss of CD47 function modulated DNA damage response and improved DNA repair in non-malignant cells also exists[28]. Thus, the evidence suggests that CD47 expression is an important factor influencing cellular response to IR and phagocytic clearance. However, the relationship between IR, DNA damage and CD47 expression, has not been studied yet.

Exposure to ionizing radiation induces DNA damage, specifically DNA double-strand breaks (DSBs)[32,33]. The Mre11–Rad50–Nbs1 (MRN) and the Ku heterodimer (Ku70-Ku80) complexes are among the first responders that bind the DSBs independently of other factors. These complexes play important roles in sensing, as well as promoting the repair of DNA ends via both HR and NHEJ pathways[34–37]. MRN and Ku70-80 complexes also have a key role in activating kinases ataxia-telangiectasia mutated (ATM) and DNA-dependent protein kinase catalytic subunit (DNA-PKcs), respectively[38–40]. In turn, these phosphoinositide-3-kinase (PI3K) family members target other proteins including the histone H2AX (referred to as γH2AX) that is phosphorylated over a large region surrounding the DSB, forming readily visualized foci, and creating a binding site for other DDR proteins[41].

In the present study, we demonstrate that IR as well as various other genotoxic agents to non-malignant and malignant cells induced elevated expression of CD47 in a dose-dependent manner. This upregulation correlates with the extent of residual DSB as determined by γH2AX staining, that is used as a marker to the extent of DNA damage caused by irradiation or other genotoxic agents. Interestingly, cells lacking mre-11, a component of the MRN complex that plays a central role in DSB repair, fail to elevate the expression of CD47 upon DNA damage induction. On the other hand, p53, that directs cellular response to DNA damage, is not required for this upregulation, which occurred in p53$^{-/-}$ cell line and was not affected by MDM-2 inhibitor. In addition, NFkB pathway or cell-cycle arrest does not account for the increase of CD47 expression. Hence, our data suggest that CD47 is upregulated following DNA damage in an mre-11 dependent-manner in both malignant and non-malignant cells.

## Results

To characterize the effect of IR on CD47 expression, we analyzed the dose and time dependency of CD47 surface expression following γ-irradiation. Using Annexin/PI staining we first confirmed cell's viability following the various doses of γ-irradiation (Fig. S1). CD47 expression on live cells, was significantly elevated on HEK-293T cells in response to γ-irradiation in a dose-dependent manner (Fig. 1a, b). Elevated expression was evident at 6 h following irradiation and peaked at 24 h (Fig. 1c). On the other hand, CD47 expression was not upregulated by other cell stress conditions, including heat shock, or incubation in serum-free medium (Fig. S2).

Surprisingly, in transfected CHO cells that express the human CD47 (CHO_CD47) under the CMV promoter, human CD47 expression was also increased following γ-irradiation (Fig. 1d). As CD47 expression on CHO_CD47 widely varied between cells (Fig. 1e), we sorted single cells expressing high levels of human CD47. The resultant clones expressed a high and relatively uniform level of CD47 that nevertheless increased following γ-irradiation (Fig. 1e).

In order to determine how γ-irradiation upregulated CD47 protein levels, we first examined if it stabilizes CD47 on the cell surface. To that end, we assessed CD47 protein's half-life by treating the HEK-293T cells with cycloheximide, a protein synthesis inhibitor. Surprisingly, when compared to control cells, γ-irradiation actually shortened CD47 half-life on the cell surface (Fig. 2a). In addition, as shown by cell-surface and intracellular staining and flow cytometry (Fig. 2b), total CD47 protein level expression was significantly elevated upon γ-irradiation and not only at the cell surface. Next, we examined whether γ-irradiation elevates CD47 mRNA expression. Indeed, CD47 mRNA levels were significantly increased in irradiated HEK-293T cells (Fig. 2d), but not in CHO cells transfected with CD47 under CMV promoter (Fig. 2e). These results indicate that γ-irradiation-induced CD47 upregulation is dependent on both transcriptional and translational processes.

Exposure to ionizing radiation induces DNA damage, the most dangerous type being DNA double-strand breaks (DSBs), in a dose-dependent manner[32,33]. Therefore, we assessed in parallel CD47 surface expression and phosphorylated nuclear histone H2AX (γH2AX), an established sensitive marker of DSBs, following γ-irradiation. While CD47 expression increased with time following irradiation (Fig. 1c), γH2AX level peaked, as expected, immediately after irradiation and decreased later (Fig. 3a, b). 24 h

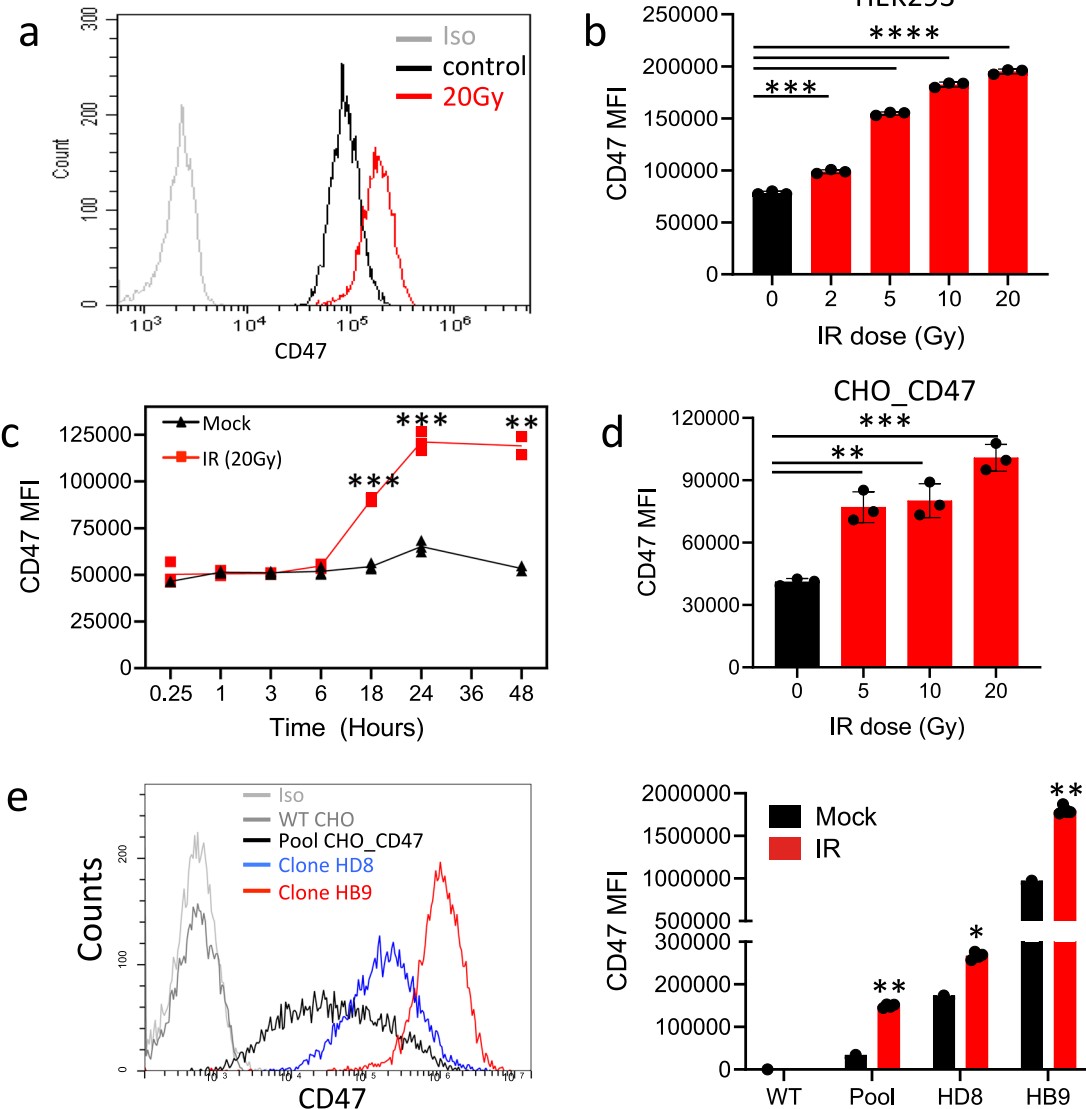

**Fig. 1 γ-irradiation induces CD47 overexpression. a** Representative histogram of CD47 staining and flow cytometric analysis before and after γ-irradiation (HEK-293T cells). **b** Dose-dependent expression of CD47 (MFI) in HEK-293T cells 24 h after γ-irradiation. **c** Time-dependent expression of CD47 (MFI) in HEK-293T cells following 20Gy γ-irradiation. **d** CD47 expression (MFI) in CHO cells transfected with human CD47 24 h after γ-irradiation.
**e** Representative histogram of CD47 staining of CHO_CD47 of different clones (left panel) and their corresponding CD47 expression (MFI) in response to 20 Gy γ-irradiation (right panel). Graphs show an average (±STD) of four replicates in each group. One of at least four independent experiments is shown. *$p < 0.05$; **$p < 0.005$; ***$p < 0.0005$.

after irradiation CD47 surface and γH2AX expression increased in parallel as seen by flow cytometry (Fig. 3c) or using western blotting of whole cell lysate (Fig. 3d), reinforcing that this increase in CD47 expression is not a result of re-distribution to the cell surface from internal pools but an absolute increase in CD47 protein expression, as seen in Fig. 2b. This increase in CD47 expression 24 h after irradiation displayed a linear correlation with the residual γH2AX levels (Fig. 3c–e), suggesting a link between the extent of residual DNA damage and increase in CD47. We further assessed whether upregulation of CD47 following DNA damage persist over time. For that end we followed expressions 1,4 and 9 days after γ-irradiation. The expression of CD47 peaked at day 1 and then gradually decreased. This transient upregulation in CD47 expression correlated with the level of residual γH2AX staining over time (Fig. 3f).

Importantly, other genotoxic agents (peroxide, Ara-C and Cisplatin) similarly induced CD47 expression in a dose-

dependent manner that correlated with the level of residual γH2AX (Fig. 3g), suggesting generalization of these findings beyond irradiation to other DNA damaging agents.

Taken together, these data clearly demonstrate that various genotoxic agents induce elevated expression of CD47 in a dose-dependent manner that correlates with the extent of double-strand break (DSB) as determined by γH2AX staining.

Several forms of DNA damage have been shown to activate p53, including those generated by IR. Shortly after DNA damage, p53 is phosphorylated resulting in enhanced stability and activity of p53 that leads to the transactivation of several genes whose products trigger cell-cycle arrest, apoptosis, or DNA repair, (reviewed in ref. [42]). Interestingly, our data suggest that p53 does not play a role in CD47 upregulation in response γ-irradiation since it also occurs in p53 mutated cell line, Huh7, that express Y220C-mutated p53 that is transcriptionally inactive[43–45] (Fig. S3A). In addition, HEK-293T cells treated with Nutlin-3, an

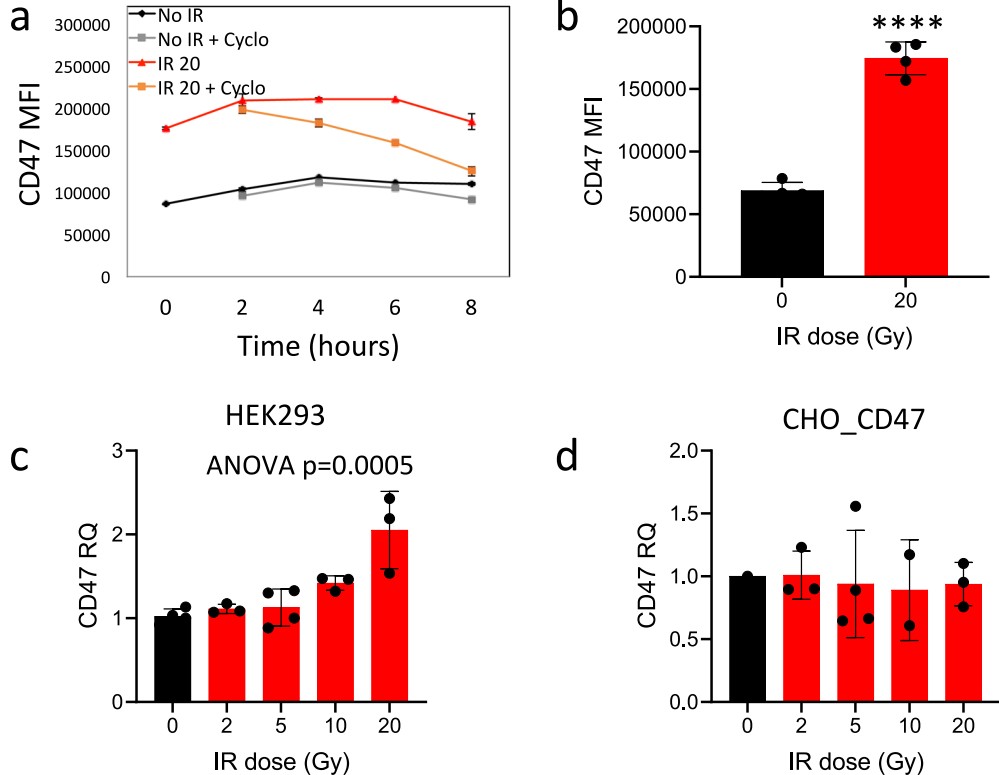

**Fig. 2 γ-irradiation regulates CD47 overexpression at the transcriptional and translational levels. a** 24 h after γ-irradiation (20 Gy) HEK-293T cells were treated with 2.5 µM cycloheximide for the indicated time points (0–8 h) and then CD47 surface expression was determined by Flow cytometric analysis. **b** CD47 cell-surface and intracellular staining (MFI) in HEK-293T cells 24 h after γ-irradiation. Graphs show an average (±STD) of four replicates in each group. **c**, **d** qRT-PCR analysis for CD47 expression 24 h after γ-irradiation in HEK-293T cells (**c**) and CHO_CD47 cells (**d**) following γ-irradiation. Graphs show an average (± STD) of three replicates in each group. One of at least three independent experiments is shown. **$p < 0.005$.

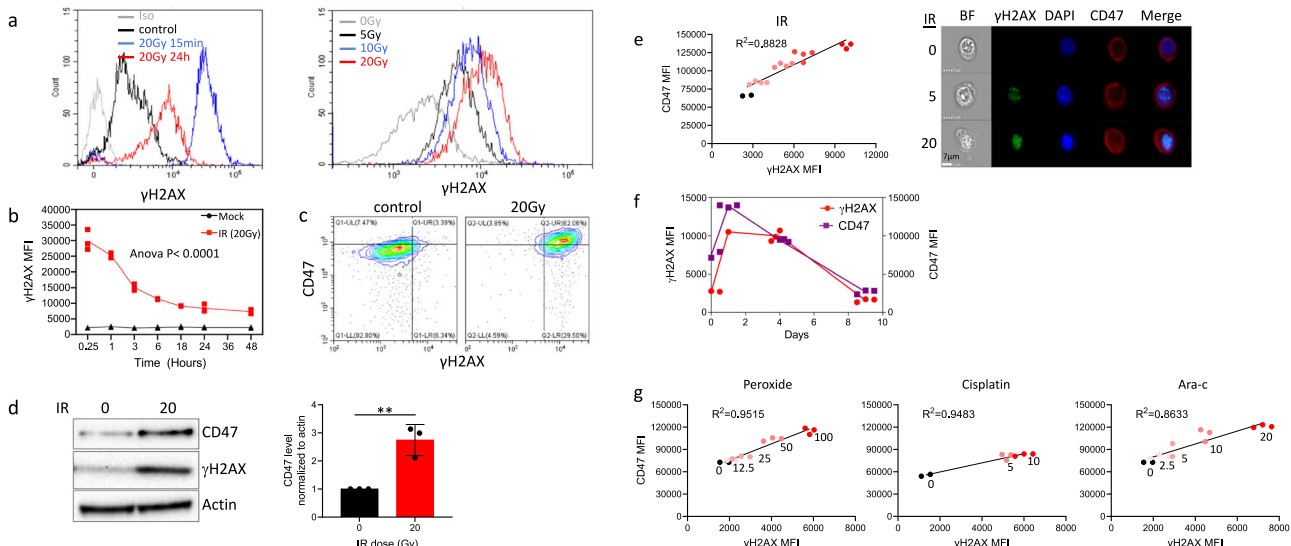

**Fig. 3 CD47 expression correlates with the level of γH2AX in response to treatment with genotoxic agents. a** Representative histograms of γH2AX staining before and after γ-irradiation. **b** Time-dependent expression of γH2AX (MFI) following γ-irradiation (20 Gy). **c** Representative dot plot of γH2AX and CD47 co-staining before and after γ-irradiation. **d** Cells were treated as above and after 24 h lysed. Western blot analyses were performed on whole cell extracts using antibodies to CD47 and γH2AX. Anti-β-actin immunoblotting revealed relative amounts of protein in each lane. Graph on the right shows an average (±STD) level of CD47 expression in triplicate samples normalized to actin in each sample (right panels). **e** The level of CD47 expression in response to γ-irradiation correlates with the level of residual γH2AX (left panel), and representative pictures taken by ImageStream of cells co-stained for CD47 and γH2AX before (0) and after γ-irradiation (5 and 20 Gy). **f** Graph depicts CD47 expression and γH2AX level (MFI) in HEK-293T cells 1,4 and 9 days after γ-irradiation. **g** The level of CD47 expression in response to different genotoxic agents correlates with the level of residual γH2AX. Graph shows an average (±STD) of four replicates in each group. One of at least three independent experiments is shown.

inhibitor of p53-MDM2 that leads to p53 activation, expressed the same levels of CD47 before and after γ-radiation as the non-treated cells (Fig. S3B).

It has been reported that NF-κB was activated in response to a variety of DNA lesions[46] and along with p53, NF-κB modulates transactivation of many genes, participating in various cellular processes involved in DDR (reviewed in ref. [47]). Treating HEK-293T cells with the IKK-2 inhibitor, SC514, that blocks NF-κB-dependent gene expression had no effect on CD47 expression before and after γ-radiation (Fig. S3C). Collectively, these data suggest that both p53 and NF-κB do not play a role in the upregulation of immune checkpoint proteins upon DNA damage.

Cells respond to DNA damage by activating cell-cycle checkpoints that transiently delay cell-cycle progression until damage is repaired (reviewed in refs. [48,49]). We posited that cell-cycle arrest might account for the increase of CD47 expression upon γ-irradiation. Therefore, we tested whether cell-cycle arrest may in and of itself induce an increase in CD47 expression. Cell cycle arrest induced by the cyclin-dependent kinase (CDK) inhibitors, isethionate (PD0332991; cause arrest at the $G_1$), and roscovitine (cause arrest at the $G_1/S$ and $G_2/M$) did not result in a significant increase in CD47 expression (Fig. S4). Thus, it is unlikely that CD47 upregulation is an indirect consequence of DNA damage-induced cell-cycle arrest.

To determine the effect of DDR pathways on irradiation-induced immune checkpoints expression, we have used pharmacological inhibitors targeting MRN, ATM and DNA-PK. Notably, mirin, an inhibitor of mre-11 exonuclease activity and MRN complex[50,51], prevented the specific irradiation-induced CD47 expression (Fig. 4a). In contrast, treatment with the inhibitors blocking either ataxia-telangiectasia mutated (ATM) (Ku55933) or DNA-PK (Nu7026) had no significant effect (Fig. 4a). We further used shRNA to silence mre-11 in HEK-293T cells that led to significant decrease in protein levels of mre-11. Interestingly, depletion of mre-11, but not shControl, prevented irradiation-induced CD47 overexpression (Fig. 4b).

Next, we used human fibroblasts derived from a patient with ataxia-telangiectasia-like disorder (ATLD2; kindly provided by Y. Shiloh from Tel Aviv University) that carry hypomorphic mutations in mre-11[52] and hence cannot assemble an active MRN complex in response to DNA damage. In contrast to normal human fibroblasts (HF) or the fibroblasts cell line BJ-5 that like ATLD2 are transformed by the human telomerase reverse transcriptase (hTERT), used as control, upon γ-radiation CD47 expression was not significantly changed in ATLD2 cells (Fig. 4c), suggesting a role to mre-11 and/or the MRN complex in this process.

To test the correlation between DNA damage and CD47 expression in vivo, we used two experimental systems. First, we induced DNA damage in mouse livers by a single injection of diethylnitrosamine (DEN). DEN is a DNA alkylating agent that leads to the formation of mutagenic DNA adducts following metabolic activation and generation of ROS mainly in zone 3 hepatocytes expressing CYP2E1[53] and mainly in these hepatocytes DNA damage is detected[54]. We performed immuno-fluorescence co-staining for CD47 and γH2AX in liver tissue sections 48 h after DEN injection, as we previously demonstrated that γH2AX staining peaked at this time point[54–56]. Remarkably, CD47 expression level was markedly increased in DEN treated livers as compared to control livers and at areas positive to γH2AX as compared to undamaged areas of DEN treated livers (Fig. 5a).

To study the expression of CD47 in response to DNA damage in tumors, we used AB12 mesothelioma cells. In vitro induction of DNA damage in AB12 cells by either γ-radiation or cisplatin resulted in significant increase in CD47 surface expression that positively correlated with residual γH2AX levels (Fig. 5b). Next, we injected $1 \times 10^5$ AB12 cells to the peritoneum of BALB/c mice to generate tumors (>95% inoculation success rates). On day 3, following tumor inoculations mice were randomly assigned to receive PBS or single injection of cisplatin (5 mg/kg), the gold standard in the treatment of mesothelioma. Six days later, the mice were sacrificed, tumors were harvested, and paraffin-embedded sections were immunostained for CD47 and γH2AX, as above. In tumor cells from cisplatin-treated mice a significant higher number of γH2AX foci were seen when compared to tumors excised from non-treated mice. Notably, CD47 expression level was also increased in cisplatin-treated tumors and positively

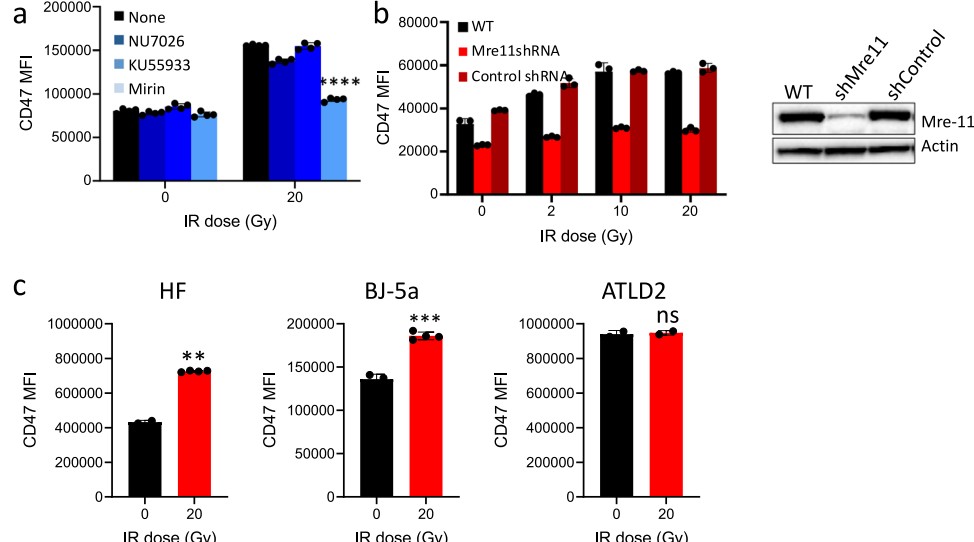

**Fig. 4 Mre-11 dependent overexpression of CD47 in response to γ-irradiation. a** HEK-293T cells were treated either with Nu7026 (10 μM), Ku55933 (10 μM), or mirin (100 μM) and then exposed to γ-irradiation. Graphs depict CD47 expression (MFI) in HEK-293T cells at 24 h after irradiation. **b** Depletion of mre-11 by shRNA in 293T cells (right panel) abrogated CD47 upregulation upon γ-irradiation (left graph). **c** CD47 expression on HF, BJ-5a and ATLD2 cells in response to γ-irradiation. Graph shows an average (± STD) of four replicates in each group. Representative of three independent experiments is shown. ns: non-significant; **$p < 0.005$; ****$p < 0.0001$.

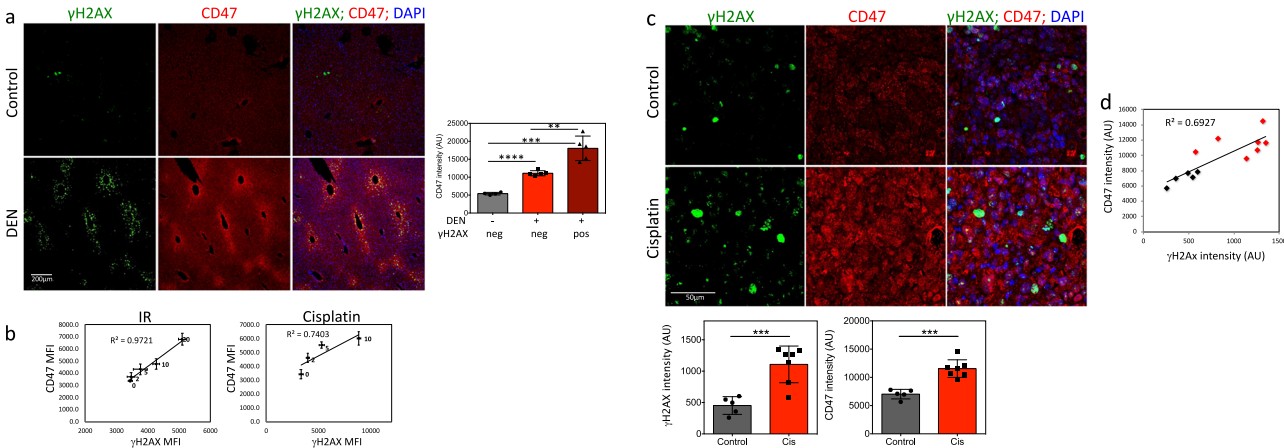

**Fig. 5 CD47 overexpression in response to DNA damage in vivo. a** Representative immunofluorescence images showing γH2AX foci (green); CD47 (red) with DAPI counterstain (blue), in control and DEN treated livers. Graph on the right shows CD47 staining intensities in control livers (gray bar; n = 5) or DEN treated livers (red bars; n = 7). In DEN treated cells' CD47 expression was measured in areas negative to γH2AX staining (light red) as compared to areas positive to γH2AX staining (dark red). **b** Dose-dependent expression of CD47 in correlation with residual γH2AX (MFI) in AB12 cells 24 h after γ-irradiation or cisplatin treatment. **c** Mice were injected with the mesothelioma AB12 cells and 3 days later tumors were either left untreated (n = 5) or were treated with cisplatin (5 mg/kg; n = 7). Tumors were harvested 6 days later. Representative immunofluorescence images showing γH2AX foci (green), CD47 (red) with DAPI counterstain (blue). Graphs below show γH2AX (left panel) and CD47 staining intensities (right panel) in control (black bars) and cisplatin-treated tumors (red bars). **d** The level of CD47 expression correlates with the level of γH2AX in both control (black dots) and cisplatin-treated mesothelioma tumors (red dots). **p < 0.005; ***p < 0.0005; ****p < 0.0001.

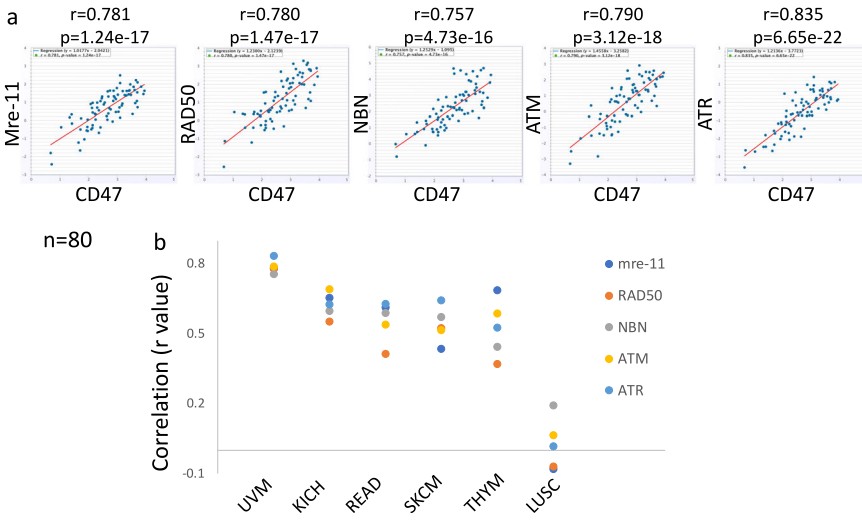

**Fig. 6 CD47 expression correlates with DDR genes in human tumors. a** Correlation between CD47 and either mre-11, ATM, ATR or RAD50 expression scores in tumor tissue of uveal melanoma (UVM) patients, is illustrated in scatter plots using StarBase. Spearman's rank-correlation coefficient r and p value are shown. **b** Summary of rank-correlation coefficients between indicated DDR gene and CD47 expression scores. Lung Squamous Cell Carcinoma (LUSC) represents tumors where CD47 and DDR genes expression do not correlate. UVM uveal melanoma, KICH kidney chromophobe, READ rectum adenocarcinoma, SKCM skin cutaneous melanoma, THYM thymoma, COAD colon adenocarcinoma.

correlated with the level of γH2AX in treated and untreated tumors (Fig. 5c, d).

Next, we investigated whether mRNAs of CD47 and DDR genes were expressed in a coordinated manner in human tumors. Using the StarBase (http://starbase.sysu) Pan-Cancer Co-Expression Analysis we found that CD47 mRNA expression positively correlated with mRNA expression of ATM, ATR, mre-11 and RAD50 in various types of cancers (Fig. 6). This positive correlation is in agreement with the findings above and all together data presented in this study demonstrates that CD47 is upregulated by genotoxic stress, via the activation of the DNA damage pathway.

We have previously demonstrated that cells expressing CD47 interact with antigen-presenting cells (APC) that in turn inhibits T cell activation[19–22]. Therefore, to test the functional immune consequences of increased CD47 expression upon DNA damage, CHO_CD47 or CHO cells were either left untreated or were irradiated and then co-cultured with peripheral blood mono-nuclear cells (PBMCs). Co-culturing PBMCs with CHO cells induced IFNγ secretion, a response that was significantly reduced when PBMCs were co-cultured with CHO_CD47 cells. Importantly, the inhibitory activity of CHO_CD47 was significantly enhanced upon irradiation (Fig. 7).

## Discussion

Immune system cells possess an ability to recognize and reject transformed cancer cells. However, tumor cells evade

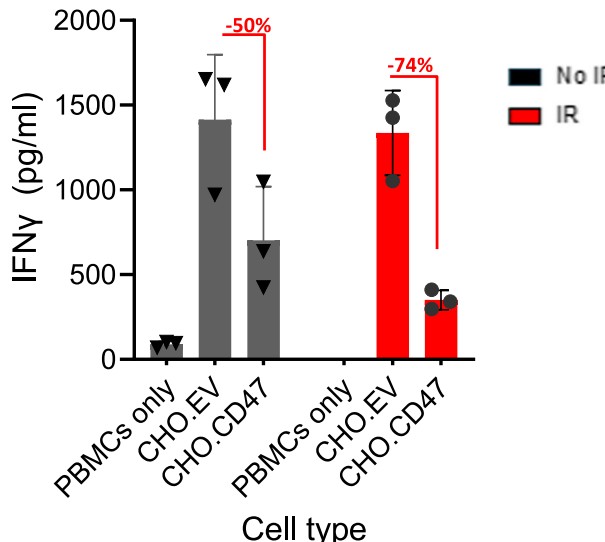

**Fig. 7 CD47-dependent T cell inhibition is enhanced upon irradiation.** CHO_EV and CHO_CD47 cells were either untreated or irradiated and were allowed to rest for 6 h. Then, cells were co-cultured with PBMCs. After 72 h, conditioned media were collected and analyzed for IFNγ secretion. The level of IFNγ secreted by PBMC cultured in the absence or presence of the CHO cells, is presented. Graph shows an average (± STD) of triplicate samples in each group. Representative of three experiments using three independent blood donors is shown.

immunosurveillance, in part through the activation of different immune checkpoint pathways that suppress antitumor immune responses. Immunotherapy relies on monoclonal antibodies that target immune checkpoints leading to cancer destruction through the activation of the host immune system. Despite the undisputable efficacy of these immunotherapies, only a fraction of patients benefits from this treatment. Studies have shown that the efficacy of immune checkpoint therapy is related to the level of IC expression, and patients having higher IC expression are the ones expected to benefit more from the respective IC inhibition therapies[57,58]. Thus, better understanding of the mechanism that controls ICs overexpression, is of great importance.

While the increase in CD47's expression following irradiation has been demonstrated in the previous study[22] the underlying mechanism for this phenomenon was not investigated. Our data clearly demonstrate that various genotoxic agents induce CD47 upregulation in a dose-dependent manner, that positively correlates with the residual level of the double strand breaks marker, γH2AX. Interestingly, we show that genotoxic stress regulates CD47 expression at both transcriptional and translational levels but do not improve and even decrease its cell surface stability.

The DNA damage response (DDR) is a complex network of genes and pathways by which DNA damage is detected and repaired. p53 is a central player in cellular DNA damage responses and activation of p53 in response to DNA damage leads and results in transcriptional activation, cell-cycle arrest, apoptosis, and DNA repair. Our data suggest DNA damage-induced CD47 upregulation is not dependent on p53, suggesting that human tumors carrying the loss of function mutations in p53 will similarly respond to DNA damage. Moreover, it is unlikely that CD47 upregulation is an indirect consequence of DNA damage-induced cell-cycle arrest, because cell-cycle arrest induced by two cyclin-dependent kinase inhibitors or serum deprivation did not trigger CD47 upregulation. It is also unlikely that CD47 upregulation is characteristic of already apoptotic cells, because these cells, significantly downregulate CD47 expression (reviewed in[26]).

Hence, the DNA damage response probably has a more direct role in the process of CD47 upregulation. The MRN (Mre-11-Rad50-Nbs1)-ATM pathway is essential for sensing and signaling from DNA double-strand breaks. Mirin is a well-known inhibitor of mre-11 exonuclease activity[50]. While mirin does not affect the ability of mre-11 to interact with Rad50 and Nbs1 or with the site of damaged DNA, it prevents MRN-dependent activation of ATM without affecting ATM kinase activity and hinders G2/M checkpoint and homology-dependent repair in mammalian cells[50,51].

Our data establish that an active mre-11 is required for the induction of CD47 overexpression upon DNA damage, as CD47 upregulation in response to γ-irradiation was abrogated by mirin treatment and did not occur in mre-11[-/-] cells. Interestingly, inhibitors to ATM or DNA-PK had no significant effect. Hence, this study is the first to show that CD47 is upregulated by genotoxic stress, via the activation of the DNA damage response pathway, and is the first attempt to study the underlying DDR-dependent mechanism.

In recent years, several studies have shown that the DDR and the immune system are tightly connected, revealing an important crosstalk between the two of them. Specifically, ligands of the NKG2D receptor, expressed by natural killer and activated CD8[+] T cells, are upregulated in response to genotoxic stress via activation of DDR pathway components[59]. In addition, a recent study demonstrated that the expression of the immune checkpoint programmed death-ligand 1 (PDL-1) in cancer cells is upregulated in response to DSBs[60]. Interestingly, the upregulation of both NKG2D ligands and PDL-1 were dependent on ATM/ATR/Chk1 kinases[59,60]. The dissociation between ATM and mre-11 found to be involved in DNA damage-induced CD47 upregulation, has to be further investigated.

Activation of SIRPα must be reliably controlled to suppress phagocytosis of viable cells when CD47 is present while allowing for efficient uptake of target cells lacking CD47. Though our data demonstrate that cells suffering from DNA damage upregulate cell surface CD47 expression, it is not clear, what is the significance of a 2-3-fold increase in CD47 expression, that is in any case, is expressed at a relative high level at normal healthy cells. A recent study has demonstrated the ratio of activating and inhibitory ligands, rather than their absolute number, regulates macrophage phagocytosis[61]. Specifically, an FcR activating antibody to CD47 ratio of 10:1 is necessary to overcome inhibition and promote phagocytosis[61]. Hence, even relatively minor increase in the number of CD47 may be an effective way for shifting the ratio toward an anti-phagocytotic outcome. In addition, we have previously shown that in addition to the anti-phagocytotic activity, CD47 also regulates macrophage maturation and function[20,21] and that this effect is also dependent on CD47 expression level[22].

One of the underlying hallmarks of cancers is their genomic instability, which is associated with a greater propensity to accumulate DNA damage. Hence, chronic activation of the DNA damage response pathway in cancer cells, augments a general phenomenon seen in non-malignant cells, and may contribute to constitutive elevated expression of CD47 in human tumors leading to immune evasion. This novel understanding of the molecular mechanism underlying CD47 increased expression should help the development of strategies to improve the efficacy of therapies using a combination of immunotherapy and the genotoxic treatments of radio and chemotherapy.

## Methods
**Cell culture, irradiation and drug treatment**. The human embryonic kidney 293T cell line (HEK-293T; ATCC; CRL-3216); The human hepatocellular carcinoma cell line Huh7; Primary normal human dermal fibroblasts (HF) were provided by the

Department of Surgery, Hadassah Hebrew-University Medical Center, Jerusalem, Israel; ATLD2 fibroblasts (a generous gift from Dr. Yosef Shiloh, Tel Aviv University, Israel). Cells were maintained in a high-glucose DMEM medium (Life Technologies, Grand Island, NY). The hTERT-immortalized fibroblasts cell line, BJ-5ta (ATCC; CRL-4001) was maintained in a high-glucose DMEM medium and Medium 199 (4:1 mixture) supplemented with 0.01 mg/ml hygromycin B. CHO cells transfected with the human CD47 were incubated in GMEM media supplemented with 0.01 mg/ml 2.5 mg/ml G418. All media were supplemented with 10% heat-inactivated fetal calf serum, 1% sodium pyruvate, and 1% penicillin/streptomycin (Biological Industries, Beit-Haemek, Israel) at 37 C and 5% $CO_2$.

Prior to γ-irradiation treatment, cells were maintained in serum-free medium and then irradiated with doses indicated in the *Results* section, using a $^{60}$Co Picker unit irradiator (1.56 Gy/min). For DNA damage induction cells were treated with either Ara-C (EBEWE pharma, Unterach, Austria), Cisplatin (Pharmachemie B.V., Haarlem, The Netherlands) or hydrogen peroxide (EMD Millipore Corp. Billerica, MA) in PBS as was previously described[54]) or were mock-treated for 30 min and were then washed with fresh media.

In addition, 10 μM ATM inhibitor (KU55933; Merck Chemicals, Burlington, MA), 10 μM mirin (Alfa Aesar, Haverhill, MA) were added at 60, 120 prior to DNA damage induction, respectively. 10 μM DNA-PK inhibitor (Nu7026; Merck Chemicals) was added 120 prior to DNA damage induction and throughout the experiment. Nutelin-3 (Sigma Aldrich, St. Louis, MO) or SC514 (CAYMAN Chemical company, Ann Arbor, MI) were added after DNA damage induction. For protein synthesis inhibition, 2.5 μM Cyclohexamide (Sigma Aldrich) was added. Roscovitine (TOCRIS bioscience, Bristol, England) and Isethionate (Selleckchem, Houston, TX) were added to cells for 24 h and then cells were collected and the level of CD47 expression was determined.

*Cytokine production.* Peripheral blood mononuclear cells (PBMC) were purified from the venous blood of healthy donors by density gradient centrifugation using Ficoll-Hystopaque (Sigma Aldrich, St Louis, MI, USA), as previously described[62]. CHO_EV (empty vector) and CHO_CD47 cells were either untreated or irradiated and were allowed to rest for 6 h. Then, cells were co-cultured with PBMCs. After 72 h conditioned media were collected and IFNγ and IL-17 levels were analyzed by ELISA (R&D Systems).

**Mre-11 shRNA knockdown.** Lentiviral vectors (pLKO.1) encoding either a non-targeting control shRNA or shRNA directed against human mre-11 pLV.MRE11i (A gift from Didier Trono (Addgene plasmid # 15565; http://n2t.net/addgene: 15565; RRID: Addgene_15565)) were cotransfected with pMD2.G and psPAX2 plasmids (Addgene 12259, 12260, respectively) into the HEK-293T packaging cell line, using Lipofectamine 2000 transfection reagent (Invitrogen) in Opti-MEM medium (Gibco). After 48 h, viral particles were harvested from the culture supernatant and filtered using syringe filter 0.45 μm pore size membrane. Viral particles, harboring either nontargeting control or mre-11 shRNA, were used to transduce HEK-293T cells. After 72 h, cells were incubated with 1ug/ml Puromycin (Sigma-Aldrich) for two weeks. Following puromycin selection, limited dilution was performed to generate single-cell clones.

**Cell lysis and immunoblotting.** Cells were lysed with X3 LDS-PAGE sample buffer (GeneScript, Piscataway, NJ). The lysates were boiled for 30 min and then $H_2O$ (1:3 v/v) was added and samples were boiled for an additional 5 min. Samples were kept at −20 ºC until use. Whole cell lysates were separated by electrophoresis on 4-20% gradient Sure-PAGE Bis-Tris gels (GeneScript) and Tris-MOPS-SDS running buffer (Gene Script) and then transferred to PVDF membranes using Trans-Blot Turbo Transfer Pack (Bio-Rad, Hercules, CA). The blots were probed with anti-CD47 (1 μg/ml; Abcam, Cambridge, England), anti-phosphorylated H2AX (1 μg/ml; γH2AX, Ser139; Bethyl, Waltham, MA), or anti-mre-11 (1 μg/ml; Novus Biologicals, Centennial, CO) followed by anti-Rabbit or anti-mouse Envision⁺ System-HRP Labeled Polymer (1:250; Agilent Dako, Santa Clara, CA) and processed with Clarity^TM Western ECL substrate (Bio-Rad). Signal was detected using the ChemiDoc^TM MP imaging system (Bio-Rad). Following stripping, the membranes were re-probed with anti-Actin mAb (MP Biomedicals, LLC, Illkirch, France). Quantitation of bands was performed using Image-Lab software (Bio-Rad).

**Flow cytometry.** Cells were trypsinized and immunostained using APC-conjugated anti-human CD47 mAb (1 μg/ml; clone CC2C6; Biolegend, San Diego, CA) for 30 min on ice. Cells were washed twice in PBS, and analyzed. CD47 expression is shown on gated live cells. For total CD47 protein level expression, CD47-stained cells (as above) were fixed and then permeabilized using FlowX FoxP3/Transcription Factor Fixation & Perm Buffer Kit (R&D Systems; Minneapolis, MN) and immunostained again with anti-human CD47 mAb. For γH2AX co-staining, CD47-stained cells were fixed and permeabilized using FlowX FoxP3/Transcription Factor Fixation & Perm Buffer Kit (R&D Systems) and immunostained with Alexa-488 conjugated anti-phospho-histone H2AX (γH2AX) mAb (1 μg/ml; clone 2F3; Biolegend). Cells were then washed and analyzed by a Cyto-FLEX Flow cytometer (Beckman Coulter, Indianapolis, USA) using the CytExpert software, and with Millipore ImageStream flow cytometer (Merck).

**RNA extraction and qRT-PCR.** Total RNA was isolated from cells using Trizol reagent (Life Technologies, Paisley, UK) and purified using MaXtract High Density (QIAGEN, Redwood City, CA). RNA yield and quantity were determined using a Nanodrop spectrophotometer ND-1000 (Thermo Scientific, Wilmington, DE). Total RNA was reverse transcribed to cDNA using oligo(dT) primers and M-MLV reverse transcriptase (Thermo Scientific). PerfeCTa SYBR Green FastMix ROX (Quanta Biosciences; Beverly, MA) was used for real-time PCR according to the manufacturer's protocol and all the samples were run in triplicate on CFX384 Touch Real-Time system c1000 thermal cycler (Bio-Rad, Hercules, CA). Cycling conditions were 95 °C for 20 s, followed by 40 cycles of 95 °C for 1 s, and 60 °C for 20 s, 65 °C for 5 s. Gene expression levels were normalized to the GAPDH gene. Primers used for qPCR analysis are: CD47 sense- TGCATTAAGGGGGTTCCTC-TACA; anti-sense- CTCTGTATTGCGGGCGTGTAT; GAPDH sense-AGCCT-CAAGATCATCAGCAATG; anti-sense-CACGATACCAAAGTTGTCATGGAT.

CD47 expression in each sample was normalized to GAPDH used as endogenous control. The data is presented and Relative quantification (RQ), relative to the untreated cells used as a reference sample.

**Animal studies.** The experimental protocol was approved by the Hebrew University Institutional Animal Care. Induction of DNA damage in vivo was performed as previously described[54–56]. Briefly, for the induction of DNA damage in vivo, C57BL/6 male mice (Harlan, Israel) were injected intraperitoneally (i.p.) with DEN (25 mg/Kg body wt; Sigma-Aldrich). Mice were sacrificed after 48 h, and liver tissues were fixed with 4% PFA.

To study the expression of CD47 in response to DNA damage in tumor tissues, we used a syngeneic orthotropic mouse malignant mesothelioma (MM) disease model[63]. $1 \times 10^5$ AB12 cells were injected to the peritoneum of BALB/c mice to generate tumors (>95% inoculation success rates). The resultant tumors are morphologically and histologically similar to human MM tumors, and respond to treatments used for MM patients, such as cisplatin. For cisplatin treatment a single injection of cisplatin (5 mg/kg) was given on day 3 following tumor inoculations, and tumor tissues were harvested from the peritoneum 6 days later.

**Immunofluorescence.** Immunofluorescence analysis of liver tissues was performed as described previously[54–56]. Briefly, livers and mesothelioma tumors were fixated in 4% formaldehyde, dehydrated, embedded in paraffin, and sectioned (5 μm-thick). To estimate DNA damage, sections were immune-stained with anti-phospho-histone H2AX (γH2AX) monoclonal antibody (4 μg/ml; Millipore, Billerica, MA). For CD47 expression, sections were co-immuno- stained with anti-CD47 (8.8 μg/ml; Abcam). Secondary antibodies Alexa-488 goat anti-mouse IgG (4 μg/ml; Molecular Probes, Eugene, OR) and Alexa-647 donkey anti-rabbit IgG (6 μg/ml; Jackson ImmunoResearch, West Grove, PA) were used. Nuclei were counterstained with DAPI (Calbiochem, Darmstadt, Germany). An Olympus BX61 microscope (Olympus, Tokyo, Japan) was used for low field image acquisition and a laser scanning confocal microscope system (FluoView-1000; Olympus) with a 10X (livers) or 40X (mesothelioma tumors) UPLAN-SApo objective for acquisition. Image quantitation was performed using ImageJ software.

**StarBase analysis.** For Bioinformatic analyses the online bioinformatics database StarBase 2.0 (http://starbase.sysu.edu.cn/) was used. StarBase is an interactive web implementation that provide visualization and analysis of large-scale data sets, designed to systematically decode interaction networks[64].

**Statistics and reproducibility.** All data were subjected to statistical analysis using the Excel software package (Microsoft) or GraphPad Prism6 (GraphPad Software Inc., La Jolla, CA, USA). A two-tailed Student's $t$ test was used to determine the difference between the groups. Data are given as mean ± SD and are shown as error bars for all experiments. Differences were considered significant at $P < 0.05$. Each experiment was repeated in at least three independent experiments.

**Reporting summary.** Further information on research design is available in the Nature Portfolio Reporting Summary linked to this article.

## Data availability
The data generated or analyzed during this study are available in Supplementary Data 1 (excel files) and in Fig. S5 (Western blots).

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

## Acknowledgements

This work was supported by grants from the Israel Cancer Association, the DKFZ-MOST Foundation, the Israel Science Foundation (ISF grant No. 1728/20) and by Hadassah-France Association, and by a research grant from KAHR Medical LTD to M.D.E. A.C is employed by KAHR Medical LTD and has options, and M.D.E has a research fund, consultation fees and options from KAHR Medical LTD.

## Author contributions

L.G., Y.V., R.H., and T.F. performed the experiments. O.W. and E.S. performed the mesothelioma experiments. A.C. and E.B. provided transfected cells. M.D.E. and J.R. designed the study and wrote the manuscript with assistance from all authors.

## Competing interests

Ayelet Chajut is employed by KAHR Medical LTD and has options, and Michal Dranitzki Elhalel has a research fund, consultation fees, and options from KAHR Medical LTD. The remaining authors declare no competing interests.
