## [Peer Review File · Communications Biology]

Reviewers' comments:

Reviewer #1 (Remarks to the Author):

The manuscript of Lucy Ghantous and collaborators focuses on the role of the DNA damage response pathway in the regulation of CD47 expression. This study is particularly interesting because CD47 is an immune checkpoint that acts as a "don't eat-me" signal to prevent phagocytosis of viable cells, and more importantly, can impede the clearance of tumor cells by macrophages.

The phagocytosis-inhibiting function of CD47 decreases at the onset of apoptosis due to various treatments used in cancer immune therapies, e.g., drugs or irradiation (IR) that induce DNA damage. However, such treatments that trigger the DNA repair machinery may also ultimately increase CD47 expression.

The link between the level of CD47 and the DNA repair process has already been suggested but the relationship between IR, DNA damage/repair, and CD47 expression is not deciphered.

Mainly, using cell lines and inhibitors, the authors provide evidence for the involvement of mre-11, a member of the MRN complex involved in the DNA damage response (DDR) pathway machinery. They showed that CD47 expression is regulated at the transcriptional and translational level and not linked to P53, a central player that directs the cellular response to DNA damage, nor to the NFκB pathway. They showed that CD47 expression is regulated at the transcriptional and translational level and not linked to P53, a central player that directs the cellular response to DNA damage.

These new findings suggest that the regulation of CD47 expression is somehow related to the DNA repair machinery and helps to understand the cellular effects of anti-cancer treatment, including cell resistance, and thus should therefore contribute to the development of immune anti-cancer therapies.

The study is well introduced and documented, experimental strategy is well adapted but additional experiments may be necessary to increase the scope of this manuscript, in particular by giving an idea of the effects of the CD47 increase.

My major concerns and questions are as follows:

In addition to some experiences that need to be clarified (to consolidate the authors' interpretation and to convince the reader), the weakness of this study lies in the absence of data on the functional consequences due to the regulation of CD47 during treatment. Measurements of apoptosis or phagocytosis assays would allow us to consider the consequences on cellular and immune responses (as mentioned below), which is not yet the case in the current form of the manuscript.

1- In the experiment using cycloheximide to inhibit protein synthesis (figure 2A), I would be more confident if another protein control should be added (e.g. GAPDH? or a membrane protein) to show if the decrease is not a general feature due to IR or is truly specific to CD47.

2- In figure 2 B, I wonder what is detected by flow cytometry after cell permeabilization, intracellular CD47 or intracellular plus cell surface CD47? The description of the staining/permeabilization procedure should be clarified, and the text (including MM) and conclusions accordingly.

3- Given that the don't eat me function of CD47 is known to be prevented by apoptosis and that IR such as other anti-cancer treatments that induce DNA damage trigger apoptosis, I think it is necessary at least, to analyze the apoptotic status of the cells after IR, in parallel with the measurement of the level and stability of CD47 level.

For example: Is this related to a delay (inhibition?) of apoptosis during DNA repair?

AnnexinV/PI assays by FACS (but other strategies are possible) could easily be done under various treatments described in the study, including mre-11 inhibition.

4- To go further, phagocytosis assay using as macrophage target, cells at various timepoint after irradiation, correlated to CD47 surface exposure, could be very informative.

Minor point the meaning of hTERT on page 10 needs to be made explicit.

Reviewer #2 (Remarks to the Author):

The paper titled "The DNA Damage response pathway regulates the expression of the immune checkpoint inhibitor CD47" by Savad et al. aims at finding a relationship between the activation of DNA repair response pathways by ionizing radiation and other cytotoxic treatments as a rheostat of CD47 expression. The authors claim that "this study is the first to show that CD47 is upregulated by genotoxic stress, via the activation of the DNA damage response pathway, and the first attempt to study the underlying DDR-dependent mechanism"; this is not entirely accurate. The authors acknowledge that others have shown that CD47 expression is upregulated after genotoxic stress such as radiation and chemotherapy (Roberts DD -which the authors cite; Semenza et al.-Not cited). Furthermore, the Roberts DD group has shown already a relationship between ionizing radiation γ H2AX and that this process is mediated by the protein Schlafen-11. The only novel aspect presented is that the mechanism of CD47 upregulation is by mre-11. This is still casual, as is only shown in mre-11-/- cells. In order to refine the mechanism, the authors should determine the role of mre-11 in the transcription and translation of CD47. Another aspect is that the authors seem to jump from different cell types using different strategies to induced genotoxic stress so there is a lack of cohesiveness. Furthermore, the authors should demonstrate the cellular phenotypic consequences of this regulation as most of the arguments made in the discussion are speculative. Another aspect is that this study solely rests on in vitro work and lacks validation using in vivo models. Thus, the publication of this study would be premature as the conclusions made by the authors are not supported by the results presented.

Minor:

- CD47 is not a "bonafide" immune checkpoint inhibitor. Thus, I suggest removing it from the title or at least adding "innate immune checkpoint..."
- The discussion arguments are very speculative and tend to focus on immune regulation, which is not what the manuscript is about.

Reviewer #3 (Remarks to the Author):

Ghantous and colleagues present an interesting new concept suggesting that radiation-induced CD47 expression is intricately linked to DNA damage and the repair complex MRN. They were able to demonstrate a correlation between these endpoints that exhibit radiation dose- and time-dependent character and the manuscript is well written. The data presented certainly warrant further consideration but there are several concerns that need to be addressed before this can be shared with a wider audience:

- Cell death under the conditions is a major concern which was not taken into consideration but could have affected the results. 20Gy and even 10Gy are large doses of radiation and many cells will have likely died or at least entered the process of undergoing cell death and therefore may have affected the staining. In fact, in Figure 3D the 20Gy-irradiated cells don't look healthy at all. It is imperative, that the authors demonstrate that the results still hold up even after exclusion of dead cells, i.e. include a viability stain for flow cytometry. Also, a FSC/SSC plot of irradiated and non-irradiated cells should be shown at least once for completion.
- Most protein data were obtained from a single cell line (HEK 293T) with a single technique (flow cytometry). Many in the audience will wonder how broadly applicable this is, hence other cell lines or at least a different technique such as Western blotting would be customary to confirm the findings.
- There is a sharp rise in CD47 MFI after 20Gy even at 0h that doesn't get mentioned or explained at all (Figure 2A red vs black line). How would the authors explain that immediate jump?
- Figure 1B and D: please add individual p values to graph bars.

- Figure 2C: at what time point after exposure was the mRNA extracted? Also, the y-axis label is insufficient. What is RQ? How were values calculated and normalized with GAPDH values?
- Figure 3F: What radiation dose was used? Viability?
- MFI units vary widely between the graphs and often they go beyond 100000. Is this correct? Traditionally, most flow cytometers have a dynamic range of about 4 decades so having MFI units beyond 100000 is somewhat unusual unless your machine is different. Can you confirm?
- The reason for choosing Ab12 cells is not really clear. Why did you decide to work with these? Wouldn't it have been useful (and straight forward) to also test these cells in vitro to confirm what was observed in the HEK293T cells? Can MM tumors be included in the human data base analysis for consistency in the story?
- Figure 5B: it is difficult to see the correlation/overlap between gH2AX and CD45. They are vastly different in quantity and location. These data are not really convincing.
- The data in Figure 6 are nice and supportive overall apart from the ATM part that is contradictive to the results from the ATM inhibitor which had no effect on the CD47 inducibility (Figure 4A). That needs to be discussed.
- The method section is not great. There is no information about the radiation device: what machine/source was used; how was radiation dosimetry performed etc.; the calculations for mRNA levels and the normalization are not described; the StarBase analysis section lacks information entirely re how data was handled/analyzed.
- Please check your figure legends. They are not very informative/complete.

RE: Manuscript COMMSBIO-22-1779-T

"THE DNA DAMAGE RESPONSE PATHWAY REGULATES THE EXPRESSION OF THE IMMUNE CHECKPOINT CD47"

Reviewer #1:

In his opening comment the reviewer states that the weakness of this study lies in the absence of data on the functional consequences. In the revised manuscript we have now added new data on AnnexinV/PI staining of cells before and after irradiation as well as functional consequences of CD47 upregulation in the context of cellular immune responses.

Specific comments:

1. The reviewer asked us whether the decrease in CD47 stability on the membrane is a general feature of IR or is it specific to CD47. In response, although we cannot say it is unique to CD47, this is not a general feature since Li Y et al., reported that irradiation and DNA damage stabilizes the TGF- β receptor on the cell surface (PMID: 31315051), suggesting that different surface receptors respond differently. In addition, the main conclusion from this experiment, whether it is unique or general phenomenon, is that increased stability of the CD47 on the surface cannot explain the increase in expression observed following DNA damage.

2. The reviewer asked us to better describe the intracellular staining procedure of figure 2B. In response, we have corrected the figure legend, text, and Materials and Methods accordingly – page 20, second paragraph, in red.

3-4. In his opening comment and in this specific concern the reviewer raises the concern regarding the absence of data on the functional consequences such as cell apoptosis following IR. In response, we now added data on Annexin/PI analysis following irradiation in supplementary figure S1. As can be seen, the percentage of apoptotic cells is negligible, and these cells indeed express reduced levels of CD47. In addition, it is important to note that all the CD47 expression data presented in this study were on gated live cells only.

5. The reviewer further suggest that it will be useful to correlate the level of CD47 surface expression at various experimental conditions to their phagocytosis by macrophages. In response, the basic CD47 expression level on these cells is quite high. Therefore, we do not expect (and did not observe previously) significant phagocytosis of these cells by macrophages, even more so after irradiation. Therefore, this kind of assay is not very informative in the current setting. However, we previously demonstrated that in addition to blocking macrophage phagocytosis, CD47 also attenuate T cell responses. Hence, in response to the reviewer comment we now added data in figure 7 that clearly demonstrate that CHO cells expressing human CD47 exposed to irradiation inhibits T cells more efficiently as compared to control cells that were not exposed to irradiation. It is presented on page 14, last paragraph and figure

7, and in the Methods section page 19 first paragraph.

Minor point:

The meaning of hTERT on page 10 was specified – page 10 last paragraph.

Reviewer #2:

Major points:

1. The reviewer states that the mre-11 link is still casual as it was only shown in cells lacking mre-11. In response, we have now validated these findings using ShRNA against mre-11 and added these data to the revised manuscript in Figure 4B and in the results section page 10 second paragraph. Importantly, data in figure 4A, where we have used the highly specific mre-11 inhibitor, Mirin, also re-inforce the importance of mre-11 in CD47 up-regulation in response to irradiation.

2. The reviewer suggest we should determine the role of mre-11 in the transcription and translation of CD47. In response, although the reviewer is right that this will strengthen the finding linking mre-11 to CD47, this is a very exploratory study not in the scope of the present manuscript, since it can be a direct or indirect effect. Nevertheless, it is important to note that mre-11's potential association with the transcriptional machinery have been proved by independent studies. Specifically, it was shown that mre-11 scans active genes for transcription to preserve the integrity of the coding genome upon DNA damage (PMID: 34020942).

3. The reviewer argues that the study is not cohesive since we use different cell types and different strategies to induce genotoxic stress. In fact, all the data was obtained with a single cell line (HEK 293T) and using γ -radiation. The use of other genotoxic stress strategies and different cell types was presented to demonstrate that the observed phenomenon is not unique to irradiation or to specific cell type, but it is a general phenomenon associated with DNA damage in many cell types.

4. The reviewer asks us to demonstrate the cellular phenotypic consequences. In response, as DNA damage may induce cell death, we now added data on Annexin/PI analysis following irradiation in supplementary figure 1A and page 5 first paragraph on the results section.

5. The reviewer states that this study rests solely on in vitro work with no validation with in vivo models. In response, as can be seen in text and figures 5A and 5C, we have used two *in vivo* models that clearly demonstrate that DNA damage induces CD47 overexpression. In addition we used in silico analysis on human tumors demonstrating a correlation between the extent of DDR genes and CD47 expression.

Minor points:

1. The reviewer claims that CD47 is not a “bonafide” immune checkpoint and suggest removing it from the title. The reviewer is right that CD47 is actually an innate immune checkpoint. However, several independent studies, including our own, have

demonstrated that blocking CD47 promotes antitumor T cell immunity and is now considered as 'immune checkpoint' by many, e.g., see 'The CD47-SIRP α Immune Checkpoint' (PMID: 32433947).

2. The reviewer claims that the Discussion tends to focus on immune regulation and not on what the manuscript is about. In response, out of 8 paragraphs in the discussion only one deals with the immune regulatory function of CD47. The rest of the discussion deals with DNA damage and the way it regulates CD47 and other immune checkpoints.

Reviewer #3:

1. The reviewer asks us to add data regarding cell death under irradiation. In response, we have now added Annexin/PI analysis following irradiation in supplementary figure S1 and page 5 first paragraph, results section. As can be seen, the percentage of apoptotic cells is negligible. In addition, the reviewer further rightly claims that it is imperative to show that the results still hold up even after exclusion of dead cells. In response, as expected, dead cells express lower levels of CD47. In addition, it is important to note that all the CD47 expression data presented in this study were on gated live cells only. This has been stressed now in the text relating to figure 1 on the results section – page 5 first paragraph and to the Materials and Methods section on page 20, second paragraph.

2. The reviewer argues that in order to make the study more relevant we have to show the effect using additional technique (like Western Blotting) and to confirm them with other cell lines in addition to HEK-293 cells. In response, the reason for using flow cytometry is that it specifically detects cell surface protein the form having functional significance. We have now added Western Blot data as figure 3D – page 7 first paragraph at the results section, to strengthen the data on CD47 expression levels in response to irradiation. As for additional cell types, the data presented in figures 4,5 (added now as 5B and results section page 11 last paragraph) show the use of several other cells representing primary, normal and cancer cells. Overall, these data demonstrate that the link between DNA damage and CD47 overexpression is a general phenomenon and not limited to a single cell type.

3. The reviewer is wondering how we can explain the sharp increase in CD47 expression at time 0 seen in figure 2A. In response, in figure 2, time 0 indicate the time of cycloheximide addition, which is 24h after irradiation, hence the sharp increase in CD47 expression. We thank the reviewer and now have corrected the figure legend to clarify this point (page 7).

4. As the reviewer suggested - individual p values were added in figures 1B and 1D.

5. The reviewer asks us few details regarding figure 2C. In response: 1) mRNA was extracted 24h after exposure to irradiation. This information was now added to the figure legend; page 7. 2) the Y-axis label RQ stands for 'Relative Quantification'. The values were calculated and normalized using the standard and accepted calculations

used for qRT-PCR. We apologize and now added to the Materials and Methods section – page 20 last paragraph.

6. The reviewer asks what radiation dose was used in figure 3F. In response, there was no use in radiation in this figure. Figure 3F describe the response to other genotoxic agents such as Cistplatin and Ara-C, and data presented is on gated live cells only.

7. The reviewer is wondering how the MFI goes beyond 100000, in some cases. In response, we confirm that the new digital FACS machines, as the one we have used, has a range of 10^7 . Furthermore, the MFI values vary in graphs describing different cell types but are quite similar when the same cell type (i.e. HEK-293T cells) are shown.

8. The reviewer wonders why we chose Ab12 cells. In response, Ab12 generate a tumor that closely recapitulate the human tumor and for which DNA damage induced by cisplatin is the gold standard treatment. In addition, there is an extensive documented expertise in our institute on mesothelioma vis a vis Dr. Ori Wald. We agree that testing the mesothelioma cells in-vitro, as suggested by the reviewer is off high value and we have now added *in vitro* data using Ab12 cells treated with cisplatin and irradiation to figure 5B and to page 11 last paragraph results section.

MM tumors were not included in the human data analysis as there are only 86 cases of MM in the data base and despite showing the same trend it does not reach the threshold statistical significance that we used.

9. The reviewer is right in his claim that the location and quantity of γ H2AX and CD47 is vastly different. The reason for that is that in contrast to the DEN model where DNA damage induction is induced at a specific zone, cisplatin-induced DNA damage is not limited to a certain area in the tumor. Consequently, both γ H2AX and CD47 expression is distributed. However, there is a clear correlated increase in the two upon induction of DNA damage by cisplatin as can be seen in D.

10. The reviewer suggests that the data in figure 6 showing a correlation between CD47 and ATM expression, contradicts the finding that ATM inhibitor has no effect on CD47 expression. However, the fact that the expression of CD47 and ATM correlates does not imply that one regulates the expression of the other, just that the two are linked via the same process or phenomenon, in this case DNA damage. The same is true for ATR. The expression of the various genes presented in figure 6 indicate that the DDR pathway is induced and the more this pathway is activated the more these tumors express CD47.

11. The reviewer asks us for more details in the Materials and Methods section. In response we have added information on radiation devise (page 18 second paragraph), qRT-PCR calculations (page 20 last paragraph) and normalization and on StarBase analysis (page 21 last paragraph) as requested by the reviewer.

Sincerely yours,

Jacob Rachmilewitz, Ph.D.

Goldyne Savad Institute of Gene Therapy,
Hadassah-Hebrew University Medical Center

REVIEWERS' COMMENTS:

Reviewer #1 (Remarks to the Author):

The paper is improved and the authors have addressed my primary concerns. They added new data, particularly Annexin V/PI analysis and "functional" experiments that demonstrate that CHO cells expressing human CD47 exposed to irradiation inhibit T cells more efficiently than control cells. The experimental procedure was also more clearly described.

Reviewer #2 (Remarks to the Author):

The authors have mostly addressed my concerns. The word "immune checkpoint" should be removed from the title as none of the results from this manuscript address this aspect of CD47.

Reviewer #3 (Remarks to the Author):

The authors addressed most of my concerns adequately. There is one remaining question:

A gating strategy must be included, this can be done as a supplementary figure. Please show an example of a forward/side scatter plot for irradiated and unirradiated cells and then how live cell gating was performed prior to CD47 staining. It is mentioned in the material and method section but it isn't clear how this was done.

Joanna Timmins, PhD
Editorial Board Member
Communications Biology

RE: Manuscript COMMSBIO-22-1779A
"THE DNA DAMAGE RESPONSE PATHWAY REGULATES THE EXPRESSION OF THE
IMMUNE CHECKPOINT CD47"

REVIEWERS' COMMENTS:

Reviewer #1 (Remarks to the Author):

The paper is improved and the authors have addressed my primary concerns. They added new data, particularly Annexin V/PI analysis and 'functional' experiments that demonstrate that CHO cells expressing human CD47 exposed to irradiation inhibit T cells more efficiently than control cells. The experimental procedure was also more clearly described.

Reviewer #2 (Remarks to the Author):

The authors have mostly addressed my concerns. The word "immune checkpoint" should be removed from the title as none of the results from this manuscript address this aspect of CD47.

In response, we have kept the words "immune checkpoint" in the title as agreed by the editor.

Reviewer #3 (Remarks to the Author):

The authors addressed most of my concerns adequately. There is one remaining question:

A gating strategy must be included, this can be done as a supplementary figure. Please show an example of a forward/side scatter plot for irradiated and unirradiated cells and then how live cell gating was performed prior to CD47 staining. It is mentioned in the material and method section but it isn't clear how this was done.

In response, we have added the live cell gating on a forward/side scatter plot for irradiated and unirradiated cells in supplementary figure 1B.

Sincerely yours,

Jacob Rachmilewitz, Ph.D.
Goldyne Savad Institute of Gene Therapy,
Hadassah-Hebrew University Medical Center